# Understanding the Relationship between Nonalcoholic Fatty Liver Disease and Thyroid Disease

**DOI:** 10.3390/ijms241914605

**Published:** 2023-09-27

**Authors:** Paulina Vidal-Cevallos, Sofía Murúa-Beltrán Gall, Misael Uribe, Norberto C. Chávez-Tapia

**Affiliations:** Obesity and Digestive Disease Unit, Medica Sur Clinic and Foundation, Av. Puente de Piedra 150, Toriello Guerra, Tlalpan, Mexico City 14050, Mexico

**Keywords:** nonalcoholic fatty liver disease, thyroid hormones, physiopathology, metabolic diseases, therapeutics

## Abstract

The prevalence of hypothyroidism in patients with nonalcoholic fatty liver disease (NAFLD) is high (22.4%). Thyroid hormones (THs) regulate many metabolic activities in the liver by promoting the export and oxidation of lipids, as well as de novo lipogenesis. They also control hepatic insulin sensitivity and suppress hepatic gluconeogenesis. Because of its importance in lipid and carbohydrate metabolism, the involvement of thyroid dysfunction in the pathogenesis of NAFLD seems plausible. The mechanisms implicated in this relationship include high thyroid-stimulating hormone (TSH) levels, low TH levels, and chronic inflammation. The activity of the TH receptor (THR)-β in response to THs is essential in the pathogenesis of hypothyroidism-induced NAFLD. Therefore, an orally active selective liver THR-β agonist, Resmetirom (MGL-3196), was developed, and has been shown to reduce liver fat content, and as a secondary end point, to improve nonalcoholic steatohepatitis. The treatment of NAFLD with THR-β agonists seems quite promising, and other agonists are currently under development and investigation. This review aims to shine a light on the pathophysiological and epidemiological evidence regarding this relationship and the effect that treatment with THs and selective liver THR-β agonists have on hepatic lipid metabolism.

## 1. Introduction

Nonalcoholic fatty liver disease (NAFLD) and thyroid disease are highly prevalent globally and are reported to be present as comorbid conditions in up to 22.4% of patients with NAFLD [1,2,3]. Primary hypothyroidism, henceforth referred to as hypothyroidism, is a disease characterized by a TSH concentration above the reference range; it can either be overt or subclinical. Overt hypothyroidism is defined by high TSH levels and free thyroxine (T4) levels below the reference range, whereas subclinical hypothyroidism is defined as TSH levels above the reference range, with free T4 levels within the population reference range [4]. The prevalence of overt hyperthyroidism ranges from 0.2% to 1.3% in iodine-sufficient countries [4], with an estimated incidence in Europe of 51 cases per 100,000 per year [5]. The estimated prevalence of overt hypothyroidism ranges from 0.2% to 5.3% in Europe and 0.3% to 3.7% in the USA [4].

NAFLD is defined by the presence of steatosis in >5% of hepatocytes on histological analysis or >5.6% when assessed using proton magnetic resonance spectroscopy or quantitative fat/water selective magnetic resonance imaging [6],.in addition to a lack of other causes of hepatic fat accumulation (alcohol, hereditary conditions, medications) [7]. The global estimated prevalence of NAFLD is 25% [8]. More recently, there has been a change in the nomenclature of this disease. A consensus was reached to use the term “metabolic dysfunction-associated steatotic liver disease” (MASLD), which requires the presence of hepatic steatosis and at least one cardio-metabolic risk factor [9]. Even though this new definition was reached, prior research work use NAFLD in their inclusion criteria; therefore, we will continue to use NAFLD throughout this paper.

The hypothalamic–pituitary–thyroid axis plays an essential role in many metabolic pathways, especially those involving lipids and carbohydrates. NAFLD has been described as the hepatic manifestation of metabolic syndrome. Therefore, a relationship between hypothyroidism and NAFLD has long been hypothesized and studied. This review aims to shine a light on the pathophysiological and epidemiological evidence regarding this relationship and on the effects of treatment, with thyroid hormones (THs) and selective liver TH receptor (THR)-β agonists, on hepatic lipid metabolism.

## 2. Pathophysiology

THs regulate many metabolic activities in several systems in the body (liver, adipose tissue, and the nervous, cardiovascular, and musculoskeletal systems) by modulating glucose and lipid metabolism. In the liver, they promote the export and oxidation of lipids, as well as de novo lipogenesis, control hepatic insulin sensitivity, and suppress hepatic gluconeogenesis. All the above-mentioned effects of THs are controlled via two mechanisms: first, by the THR acting directly on gene expression; and second, in association with nuclear receptors, such as the peroxisome proliferator-activated receptor (PPAR), liver X receptor, and bile acid signaling pathways. The THR has two major isoforms: α and β. THR-α is expressed in the brain, white adipose tissue, and myocardial atria. Brown adipose tissue contains both isoforms, and THR-β, which is the predominant isoform in the liver and heart ventricles, mediates the action of TH on TSH production in the pituitary gland [10].

Because of its importance in lipid and carbohydrate metabolism, the involvement of thyroid dysfunction in the pathogenesis of NAFLD seems plausible. Many mechanisms have been implicated in this relationship: TSH levels, low TH levels, and chronic metabolic response (Figure 1) [11].


**High TSH Levels**


High levels of serum TSHs are the hallmark biochemical alteration in hypothyroidism, whether overt or subclinical, so understanding its role in the pathogenesis of NAFLD induced by hypothyroidism is essential.

The TSH binds to its receptor on the surface of hepatocytes and stimulates the PPARα pathway and the activation of the sterol regulatory element-binding transcription factor 1-c, and in doing so, promotes hepatic lipogenesis [12,13]. Its high levels have also been associated with the decreased activity of hepatic lipoprotein lipase, thus promoting hepatic triglyceride accumulation [11,13].

The TSH increases hepatic gluconeogenesis through the regulation of gene transcription that encodes for rate-controlling enzymes. The TSH enhances hepatic cAMP-regulated transcriptional coactivator-2 expression in the liver, which in turn stimulates the expression of glucose-6-phosphatase and cytosolic phosphoenolpyruvate carboxykinase. These promote the decarboxylation of oxaloacetate to phosphoenolpyruvate, and the hydrolyzation of glucose-6-phosphate into free glucose and inorganic phosphate [14]. The alteration in the regulation of carbohydrate metabolism due to higher levels of the TSH could lead to insulin resistance and type 2 diabetes, and in doing so, could increase the risk of NAFLD. Finally, the TSH can decrease the phosphorylation of HMGCoA reductase, thereby inducing hypercholesterolemia [13].

A recent descriptive cross-sectional study, performed on 2452 subjects, found that TSH ≥ 2.5 μIU/mL is associated with a 1.5-fold increase in the risk of NAFLD and a 1.8–2.5-time increased risk of liver fibrosis, when adjusted for confounding variables. The study also found a higher prevalence of NAFLD (define by FLI score) in the population with TSH ≥ 2.5 μIU/mL when compared to TSH < 2.5 μIU/mL [15]. These data show epidemiological evidence for the relationship between NAFLD and raised levels of TSHs, and the pathophysiological summary previously presented might explain why subclinical hypothyroidism has also been epidemiologically associated with NAFLD, not requiring low levels of THs for these metabolic changes [12].


**Low Thyroid Hormones**


Low TH levels are seen in overt hypothyroidism. They normally regulate lipid metabolism in a tissue-dependent way, and are contingent on the receptor that they act on. Triiodothyronine (T3) and T4 are made by the thyroid, and their production is controlled centrally by the hypothalamus and the pituitary gland [16]. Peripherally, the activity of THs is controlled by multiple mechanisms. 

First, there is a control in the entry of THs into the target cells given by TH transport proteins, such as monocarboxylate transporter 8, monocarboxylate transporter 10, or organic anion transporting polypeptide 1c1, which have a distinct expression throughout different organs [17]. Second, deiodinases (DIO1, DIO2, and DIO3) are intracellular enzymes that catalyze the conversion of THs. T4 is converted into the active form T3 by DIO1. T3 and T4 are converted into inactive forms (rT3 and T2) by DIO3 and DIO2; this is true for all tissues [13,16,17]. 

The final step for TH action depends on the THR (both THR-α and THR-β isoforms), as mentioned previously. These mediate the genetic effects of THs [17] by direct ligand binding, which causes a conformational change in the THR and the release of a co-activator complex that results in the initiation of transcription. Another mechanism by which THs can regulate gene transcription is through their effect on transcription factors, such as forkhead box protein O, phosphoinositide 3-kinase, and αvβ3 integrin [18,19,20]. 

In particular, in a healthy liver, the main regulators of TH action are DIO1, which is highly expressed in hepatocytes, and DIO3 and THR-β, which are ubiquitous in stromal cells (fibrogenic myofibroblasts) [16,17]. The many mechanisms that control cell entry and the action of THs suggest that there are different levels of the TH circulating in every tissue at any given time. Thus, although circulating TH levels can be normal, there is a possibility for decreased levels inside specific cells or organs [17].

DIO2 is implicated in the conversion of T3 and T4 to T2. T2 has been shown to reduce adiposity and dyslipidemia, increase the lipid mobilization and secretion of VLDL by the liver, and reduce lipid droplets in the liver by promoting fatty acid oxidation [21].

This complex physiology gives room for a variety of altered mechanisms that could explain the relationship between decreased THs and NAFLD, which are discussed next. Decreased TH levels are associated with the reduced liver uptake of free fatty acids, decreased lipolysis in adipose tissue, and decreased cholesterol clearance, resulting in elevated circulating levels of low-density cholesterol (LDL), high-density cholesterol, and triglycerides [13].

THs stimulate carnitine palmitoyltransferase-1a, the rate-limiting enzyme in fatty-acid oxidation. When levels of the TH are decreased, this enzyme is downregulated and the oxidation of free fatty acids is reduced, which leads to the hepatic accumulation of triglycerides. Hepatic lipase is also diminished, which contributes to the accumulation of triglycerides in the liver [13].

The expression of HMGCoA reductase in the liver is the primary regulator of cholesterol biosynthesis. Its expression is largely impacted by THs, mainly T3. In rats, it has been shown that after T3 administration, the level of HMGCoA reductase increases, whereas the levels of intrahepatic cholesterol decrease [22]. Nevertheless, there are two mechanisms that can oversee this protective effect of hypothyroidism. First, increased intestinal cholesterol absorption occurs through the Niemann–Pick C1-like 1 protein and second, there is a decrease in the cell surface LDL receptor (LDL-R) [13]. THs induce the expression of LDL-R in hepatocytes via two pathways. The activation of the THR recruits the LDL-R promoter, and the activation of sterol regulatory element-binding transcription factor-2 activates LDL-R [22,23]. These promote hepatic LDL endocytosis and lower circulating LDL levels, which are also promoted by the reduction of protein convertase subtilisin/kexin type 9 [22,24,25].

THs also directly stimulate several enzymes that activate lipogenesis (acetyl-CoA carboxylase, fatty acid synthetase) and transcription factors that participate in de novo lipogenesis (carbohydrate responsive element-binding protein). In doing so, they avoid lipid hepatocyte accumulation [13,26]. In hepatocytes, T3 can also promote intrahepatic lipolysis via lipophagy, further reducing lipid accumulation in hepatocytes [13,16].

Some molecules act as coactivators and corepressors in TH signaling (nuclear receptor corepressor-1, nuclear receptor corepressor-2, and HDAC3). These factors also bind to nuclear receptors, such as the THR, PPAR, retinoic acid receptor, and liver X receptor. When any of these are altered or knocked out in mice, it can lead to hepatic steatosis (nuclear receptor corepressor-1) and obesity (nuclear receptor corepressor-2), thus contributing to NAFLD. These coactivators can also change the peripheral sensitivity to THs [22].

The action and activation of THR-α and THR-β by THs are essential to the pathogenesis of NAFLD induced by hypothyroidism. THR-α knockout mice have decreased lipogenesis, with decreased liver fat content and white adipose tissue. As such, these mice have less hepatic steatosis and insulin resistance. On the other hand, THR-β knockout mice have increased hepatic lipid accumulation and decreased β-oxidation, with no changes in adipose tissue [13].


**Chronic Inflammation and Hormone Interactions**


The accumulation of lipids in the liver is also increased by obesity and low resting energy expenditure (REE), which are both associated with hypothyroidism. A recent retrospective study of obese, non-diabetic women compares REE in patients with primary hypothyroidism and levothyroxine (LT4) treatment and patients with normal thyroid function. The study found that REE is decreased in patients with hypothyroidism, even when treated with LT4 and adjusted by body max index, age, body composition, and amount of physical activity. This highlights that the metabolic abnormalities in hypothyroidism go beyond T4 blood levels [27].

The accumulation of lipids in the liver is followed by a chronic inflammatory state, which can lead to nonalcoholic steatohepatitis (NASH) and fibrosis. This is mediated by hepatic insulin resistance secondary to intrahepatic triglyceride accumulation, and by reduced β-pancreatic cell sensitivity to the glucose in patients with hypothyroidism [13,16].

Oxidative stress and the production of reactive oxygen species have also been described in these patients. They are linked to elevated concentrations of free fatty acid and mitochondrial dysfunction that produce free radicals, which in turn activate proinflammatory cytokines (tumor necrosis factor-α and transforming growth factor-β) and stellate cells [11]. Reactive oxygen species have been shown to diminish deiodinase activity, which can further impair the activation of THs, thus perpetuating this cycle [16].

Patients with hypothyroidism have high levels of leptin, a hormone that regulates appetite and enhances hepatic insulin resistance by the dephosphorylation of insulin receptor substrate-1. Decreased levels of adiponectin have been found in patients with hypothyroidism, which is also associated with an increase in hepatic insulin resistance. This observation is nonetheless controversial because other studies have not been able to find an association between thyroid function and adiponectin levels [11,16,28].

## 3. Epidemiological Evidence

As previously discussed, there is plenty of pathophysiological evidence to prove that the association between NAFLD and hypothyroidism is plausible. A large effort has also been made to prove this association in epidemiological studies. In so doing, both positive and negative results have been produced. There are five systematic reviews and meta-analyses that investigate this relationship.

He et al. published a systematic review and meta-analysis that evaluated this association. They included 13 studies (cohort, case-control, and cross-sectional) from 11 countries, with 42,143 patients. They found that overt and subclinical hypothyroidism independently increase the risk of NAFLD (aOR = 1.72, 95% CI 1.32–2.23, $I^{2}$ = 75%), and this remains true when analyzed separately: overt (aOR = 1.81, 95% CI 1.30–2.52, $I^{2}$ = 36.3%) and subclinical (aOR = 1.63, 95% CI 1.19–2.24, $I^{2}$ = 80.6%) [29].

A systematic review and meta-analysis from 2017 analyzed 14 observational studies involving 7,191 patients with NAFLD and 30,003 controls. These authors did not find a significant association between NAFLD and either overt hypothyroidism (pooled OR = 1.37, 95% CI 0.78–2.41, *p* = 0.27, $I^{2}$ = 81%) or subclinical hypothyroidism (pooled OR = 0.63, 95% CI 0.18–2.20, *p* = 0.47, $I^{2}$ = 97%), or with levels of T3, T4, or TSH [30].

In 2018, Mantovani et al. published a systematic review and meta-analysis comprising 12 cross-sectional and 3 longitudinal studies that enrolled 44,140 patients. They found that hypothyroidism was associated with NAFLD (aOR = 1.42, 95% CI 1.15–1.77, $I^{2}$ = 51.2%). Using the three longitudinal studies, subclinical hypothyroidism was not independently associated with NAFLD (defined with ultrasound) over a median of 5 years (HR 1.29, 95% CI 0.89–1.86, $I^{2}$ = 83.9%) [31].

Guo et al. performed a meta-analysis of the relationship between thyroid function parameters and NAFLD and/or NASH, using 26 articles with a total of 61,548 patients. Their main findings were that regardless of age and thyroid state, increased TSH levels were significantly associated with a higher risk of NAFLD/NASH (OR = 1.605, 95% CI 1.180–2.183, *p* = 0.003, $I^{2}$ = 87.5%), and that regardless of age, unclassified hypothyroidism was associated with a risk of NAFLD/NASH (OR = 2.317, 95% CI 1.425–3.768, *p* = 0.001, $I^{2}$ = 28.4%) [32].

Because of the observational nature of the studies included in these three meta-analyses, a causal relationship could not be established. Nevertheless, in 2021, a Chinese study used a genome-wide association study and Mendelian randomization to evaluate the causality of this relationship. They obtained two genome-wide association study databases, a hypothyroidism database (10,211 samples of hypothyroidism cases and 12,211 controls), and a NAFLD database (1483 NAFLD cases and 17,781 controls). This study found an OR of 1.7578 (95% CI 1.1897–2.5970, *p* = 0.0046), proving for the first time that hypothyroidism was causally associated with an increased risk of NAFLD [33].

## 4. Effects of Hypothyroidism Treatment in NAFLD

Given the important and vast effects that THs have on lipid and carbohydrate metabolism, as well as on hepatic steatosis, it can be hypothesized that the use of exogenous THs as a treatment for hypothyroidism could improve NAFLD and the metabolic disarrays discussed above. Several studies have tried to answer this question, both in patients with overt and subclinical hypothyroidism, and even in euthyroid patients [34,35,36].

In 2017, Lui et al. performed a post hoc analysis of a randomized clinical trial to test the effect of LT4 replacement therapy in 33 patients with significant subclinical hypothyroidism (all received LT4) and 330 with mild subclinical hypothyroidism (randomized 1.5:1 to LT4 replacement therapy and no treatment). Within the significant subclinical hypothyroidism group, the prevalence of NAFLD decreased from 48.5% to 24.2% (*p* = 0.041). In the mild subclinical hypothyroidism and LT4 treatment group, NAFLD prevalence decreased from 44.2% to 35.9% (*p* = 0.108). In a subgroup analysis of patients with mild subclinical hypothyroidism and dyslipidemia, treatment with LT4 was associated with a reduction in NAFLD prevalence and serum liver enzymes [36].

In another study aimed to investigate the changes in intrahepatic lipid concentration, 20 euthyroid male Asian patients with type 2 diabetes, who received LT4 treatment titrated to achieve a TSH level of 0.34–1.7 mIU/L, showed a significant decrease in intrahepatic lipid concentration after a 16-week treatment period. The average baseline intrahepatic lipid concentration in the study population was 13%. After treatment, there was a decrease in intrahepatic lipid concentration by 2% (−2% absolute difference, 95% CI, −3 to 0; relative difference −12% ± 26%, *p* = 0.046) [34].

Even if these studies have demonstrated the benefits of the use of LT4, both in euthyroid and hypothyroid patients, exogenous THs are not innocuous, and their systemic effects can be deleterious. In patients with NAFLD and hypothyroidism, there is a clear indication for the use of LT4, and LT4 should be prescribed accordingly. Nevertheless, the overall benefit in euthyroid patients is less well understood. Thus, efforts have been made to find therapeutic targets that allow THs to exert their beneficial effects on hepatic steatosis and metabolism, without their systemic deleterious effects [13,16].

## 5. Thyroid Hormone Receptor Agonists and Metabolites

Because hypothyroidism is associated with hypercholesterolemia and atherosclerosis, investigations on the benefits of THs in the treatment of dyslipidemia have been underway for some time (Table 1).

As mentioned above, these beneficial effects are mediated by THR-β, the predominant liver thyroid hormone receptor (Figure 2). However, it is also known that excessive levels of TH can lead to adverse effects on cardiac and bone metabolism, largely mediated by the interaction with the THR-α isoform. For this reason, the investigation has been centered on finding a selective agonist for THR-β [45].

## 6. Resmetirom

Resmetirom (MGL-3196), an orally active selective liver THR-β agonist, was developed for the treatment of dyslipidemia. The first phase I study showed that it was safe and well tolerated, and it also showed the significant reduction, when compared with a placebo, of LDL (30%) and triglycerides (60%), with doses ranging from 50 to 200 mg a day [46].

More recently, a phase II trial [47] was conducted to test its efficacy for the treatment of NASH. This was a multicenter study of adults with biopsy-confirmed NASH, a hepatic fat fraction of ≥10% at baseline assessed by an MRI-proton density fat fraction, and fibrosis stages 1 to 3. Patients were assigned to Resmetirom, with 80 mg taken orally once daily or a placebo. The primary end point was total liver fat reduction at 36 weeks. They reported a reduction of fat content, with a mean difference from baseline over placebo of −28.4% (95% CI −41.3 to −15.4; *p* < 0.0001). As a secondary end point, NASH also improved on liver biopsy (27.4% Resmetirom vs. 6.5% placebo, *p* = 0.02). A nonsignificant improvement in fibrosis was also reported.

A post hoc analysis performed on quality of life demonstrated that a decrease in the proton density fat fraction of ≥30% by week 12 was independently associated with an improvement in physical functioning and physical component summary scores at week 36 (*p* < 0.05) [48].

It is important to note that neither NASH resolution nor fibrosis regression was a primary end point in the phase II trial. Currently, a phase III trial is underway (MAESTRO-NASH, NCT03900429) that plans to enroll patients with NASH and stage 2–3 fibrosis to evaluate NASH resolution as the primary end point and fibrosis regression as the secondary end point at 52 weeks. The trial is set to end in March 2024 [49].

## 7. Hyperthyroidism and Nafld

Overt hyperthyroidism is defined as low TSH concentrations with high T4 levels [4]. Hyperthyroidism seems to have a protective effect on the incidence of NAFLD. A small cross-sectional study in China found that the prevalence of NAFLD in patients with hyperthyroidism was 11.9%. Interestingly, higher T3 levels were associated with lower liver fat content (*p* = 0.009) and a decreased risk of NAFLD (OR 0.267, 95% CI 0.087–0.817, *p* = 0.021) [50]. A large case-control study in Germany confirmed these findings and found that hyperthyroidism was associated with a lower risk of NAFLD (OR 0.85, 95% CI 0.77–0.94, *p* < 0.001) [51].

## 8. Conclusions

As shown by the evidence summarized in this review, the relationship between hypothyroidism and NAFLD is not only physiologically plausible, but has also been epidemiologically proven. Despite this, there is still a long way to go to understand the nuances in the metabolic pathways that affect the hypothalamus–pituitary–thyroid axis regarding NAFLD.

Treatment with THR-β agonists also seems quite promising. We are currently awaiting the results from the Resmetirom phase III trial to know its effects on NASH and fibrosis. Another THR-β agonist, TG68, is likewise under investigation. A trial in murine models showed that treatment with TG68 for 3 weeks reduced hepatic steatosis, serum transaminases, and triglycerides levels [52]. This leaves a huge field for research and many opportunities to find further treatment options that help the growing population of individuals affected by NAFLD.

## Figures and Tables

**Figure 1 ijms-24-14605-f001:**
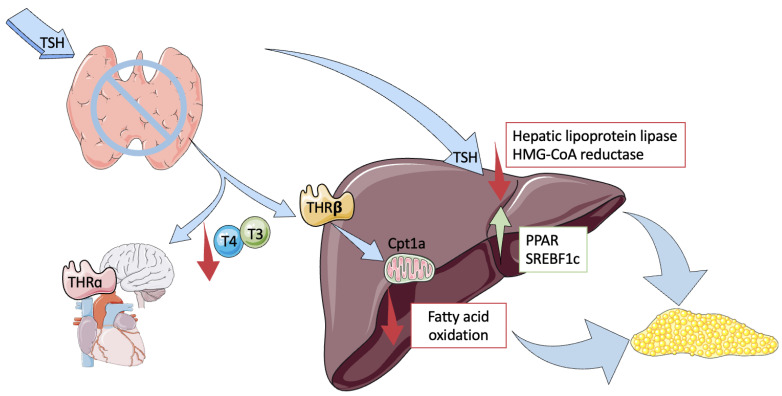
The physiological relationship between the thyroid and liver. The production of thyroid hormones (T3 and T4) is regulated by hypothalamic and pituitary activity, through the thyroid-stimulating hormone (TSH). T3 activity impacts many functions in the liver, adipose tissue, and nervous, cardiovascular, and musculoskeletal systems, by modulating glucose and lipid metabolism. These occur via thyroid hormone receptors (THRs) in different tissues. Thyroid hormone receptor-alpha (THR-α) is expressed in the brain, white adipose tissue, and myocardial atria. In the liver, TSH promotes hepatic lipogenesis using the PPAR pathway and the activation of the sterol regulatory element-binding transcription factor 1c. TSH also decreases the activity of hepatic lipoprotein lipase, thus promoting hepatic triglyceride accumulation, which increases hepatic gluconeogenesis through the regulation of gene transcription. TSH also decreases the phosphorylation of 3-hydroxy-3-methylglutaryl coenzyme A (HMGCoA) reductase, thereby inducing hypercholesterolemia. A decrease in THs is associated with the reduced liver uptake of free fatty acids, which leads to the hepatic accumulation of triglycerides, via the downregulation of carnitine palmitoyltransferase-1a (Cpt1a). Hepatic lipase is also diminished, which contributes to the accumulation of triglycerides in the liver. HMG-CoA reductase: 3-hydroxy 3-methylglutaryl-coenzyme A, PPAR: peroxisome proliferator-activated receptor, SREBF1c: sterol regulatory element-binding protein 1.

**Figure 2 ijms-24-14605-f002:**
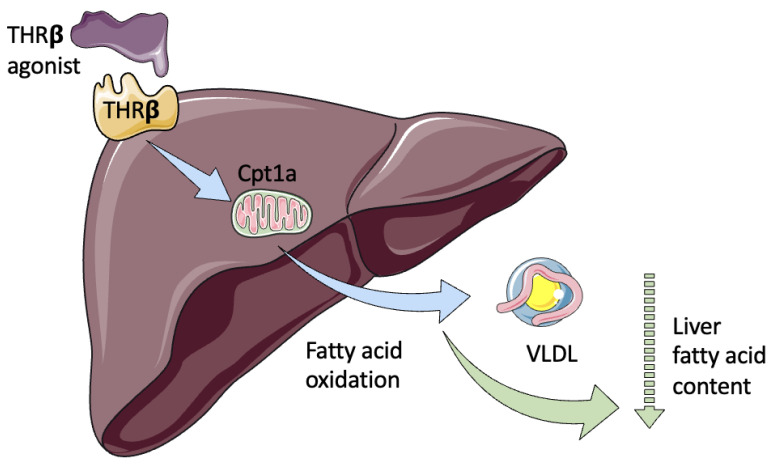
Effect of THR-β agonists in fatty liver content. THR-β can selectively activate THR in the liver to promote its antisteatotic hepatic action, without the systemic deleterious effects of exogenous thyroid hormones. Cpt1a: Carnitine palmitoyltransferase-1a, VLDL: very-low density lipoprotein.

**Table 1 ijms-24-14605-t001:** Published trials on the effects of THR agonists and TH metabolites on the treatment of NAFLD. THR thyroid hormone receptor, LDL low-density lipoprotein, TAG triglycerides, ALT alanine aminiotrasnferase, AST aspartate aminotransferase, GGT gamma-glutamyl transferase, ApoB apolipoprotein B, HDL high density lipoprotein. DIPTA: 3,5 diiodothyropropionic acid, Triac; Triiodothyroacetic acid. ↓: decrease, ↑ rise, =: no change.

Compound	Study	Study Participants	Effects on Lipids	Effects on Liver	Side Effects
THR-β receptor agonists					
Sobetirome	Grover et al. [37] Villicev et al. [38] Vatner et al. [39]	Murine models.	↓ LDL	↓ Liver fat	Stopped after phase 1. ↑ hyperglycaemia and insulin resistance.
Eprotirome	Sjouke et al. [40] Angelin et al. [41]	Patients with familial hypercholesterolaemia + statin treatment. Patients with primary hypercholesterolaemia.	↓ LDL ↓ TAG ↓ LDL ↓ Apo B = HDL	↑ ALT ↑ AST ↑ GGT ↑ ALT	Potential liver injury.
**TH Metabolites**					
Omzotirome	van der Valk et al. [42]	Patients with metabolic syndrome.	= TAG	= ALT = AST = GGT = Liver Fat	↑ FT4.
DITPA	Landenson et al. [43]	Stable congestive heart failure.	↓ LDL ↓ TAG	Not studied.	↓ TT4, TSH ↑ Heart rate Poorly tolerated.
Triac	Sherman et al. [44]	Patients with thyroidectomy and radioiodine ablation.	↓ LDL ↓ TAG = HDL	Not studied.	Increased skeletal turnover.

## Data Availability

Data sharing not applicable. No new data were created or analyzed in this study. Data sharing is not applicable to this article.

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
