# Peer review of "Understanding the Relationship between Nonalcoholic Fatty Liver Disease and Thyroid Disease"

_ijms, 2023, doi:10.3390/ijms241914605_

Round 1
Reviewer 1 Report
In the present manuscript the Authors report a narrative review on the association between NAFLD and thyroid function as well as on the impact of thyroid hormone agonists on NAFLD. The topic is certainly timely and of significant importance. I have the following comments:
1- The authors define hypothyroidism as high TSH levels. I recommend specifying that you are only talking about primary hypothyroidism, as secondary (pituitary) is associated with low TSH and low FT4 levels.
2- Consider moving from the NAFLD to the MASLD terminology, or at least mention that the terminology and diagnostic criteria are being revised.
3- When discussing the association between hypothyroidism and NAFLD I would also mention the effect related to reduced REE in hypothyroid patients (even after LT4 treatment as shown in a recent study in the setting of bariatric surgery candidates: doi: 10.1210/clinem/dgaa097)
English is fine.
Author Response
Response to Reviewer 1 Comments
In the present manuscript, the Authors report a narrative review on the association between NAFLD and thyroid function as well as on the impact of thyroid hormone agonists on NAFLD. The topic is certainly timely and of significant importance. I have the following comments:
Point 1: The authors define hypothyroidism as high TSH levels. I recommend specifying that you are only talking about primary hypothyroidism, as secondary (pituitary) is associated with low TSH and low FT4 levels.
Response 1: Thank you for pointing this out. We have further specified this in the
manuscript. See Introduction, page 1, line 29.
Point 2: Consider moving from the NAFLD to the MASLD terminology, or at least mention that the terminology and diagnostic criteria are being revised.
Response 2: Thank you for bringing the new MASLD nomenclature forward. We have specified the differences between NAFLD and MASLD and, we have pointed out that NAFLD will be used throughout the paper because it was the nomenclature used to define prior research work cited in our paper. See Introduction, page 1, paragraph 2, lines 42 -87.
Point 3: When discussing the association between hypothyroidism and NAFLD I would also mention the effect related to reduced REE in hypothyroid patients (even after LT4 treatment as shown in a recent study in the setting of bariatric surgery candidates: doi: 10.1210/clinem/dgaa097)
Response 3: Thank you for highlighting this interesting point. We have added the
suggested paper to show that not all metabolic abnormalities in primary hypothyroidism depend on T4 blood levels nor they are fully fixed by LT4 therapy. See “Chronic inflammation and hormone interactions”, page 5, paragraph 1, lines 293 - 299.
Reviewer 2 Report
The review “Understanding the Relationship between Nonalcoholic Fatty 2 Liver Disease and Thyroid Disease” is well written. Minor points:
1. The subheading discussion can be removed and replaced with Conclusions
Good
Author Response
Response to Reviewer 2 Comments
The review “Understanding the Relationship between Nonalcoholic Fatty 2 Liver Disease and Thyroid Disease” is well written. Minor points:
Point 1: The subheading discussion can be removed and replaced with Conclusions.
Response 1: Agreed. Thank you, we have replaced the title. See page 8, line 440.
Reviewer 3 Report
Vidal-Cevallos et al.’s manuscript reviewed the association of thyroid hormonal activity and NAFLD. Authors introduced that THR-beta agonist can improve the pathology of NAFLD.
Overall, it is easier to read if the abbreviation of the protein name is used in the manuscript (Ex, PPAR, RAR, HMGCoA and so on).
The manuscript needs moderate editing of English language.
Several points that I recognized will be written down.
The first sentence of both abstract and introduction is not necessary. Or is there data of the population rates of the patients with both thyroid disease and NAFLD?
Line 14~, The relationship may include high TSH levels, low TH levels and chronic inflammation.
Line15-16, Is the sentence correct? I think the authors want to mention that activity of THR-beta is effective in treating NAFLD.
Line 18-20, The sentence is not complete. Parhaps, " Because treatment of NAFLD with THR-beta agonists seems quite promising, novel other THP-beta agonists are currently under development and investigation (by pharmaceutial companies and researchers?).
Line44, What are "both diseases"?
Line46, "the effect that treatment..." might be "the effects of treatment..."?
Line53, What is "this"? Hepatic lipogenesis or gluconeogenesis?
Line 58, "ventricles," is "ventricles." ?
Figure1 is too busy to understand. I hope the authors to split into 2 figures, including an image of high TSH-low TH-chronic hepatic inflammation axis and a scheme of beneficial effects of THR-be-ta antagonist on NAFLD. And write down how to understand the Figure in legend. Line64-78, there are other contents than Figure 1.
Line107, "summery" is "summary".
From Line110, there are explanations of DIOs except of DIO2. Is the information of DIO2 rare?
Line 181-182, Is "such as diacylglycerols and ceramides" necessary here?
Line 270, "heigh" is "high-".
In Table1, It is better to unify the name of the compounds, whether it is a trade name or a generic name.
The manuscript needs moderate editing of English language. Several points that I recognized were written down as my comments.
Author Response
Response to Reviewer 3 Comments
The manuscript needs moderate editing of English language. Several points that I recognized will be written down.
Point 1: Overall, it is easier to read if the abbreviation of the protein name is used in the manuscript (Ex, PPAR, RAR, HMGCoA and so on).
Response 1: Thank you for pointing this out. We have added de following abbreviations to make the paper easier to read:
PPAR: See lines 102, 118, 134, 140, 271.
RAR: was kept as retinoic acid receptor as it is only motioned once.
HMGCoA: See lines 130, 133, 153, 252, 254.
Point 2: The first sentence of both abstract and introduction is not necessary. Or is there data of the population rates of the patients with both thyroid disease and NAFLD?
Response 2: Thank you for bringing to our attention that the initial sentenced lacked information, and therefore relevance. To fix this, we added the prevalence of hypothyroidism in patients with NAFLD, as to shed a light on the importance of the relationship between these two entities. See page 1, lines: 9-10, and 28-29.
Point 3: Line 14~, The relationship may include high TSH levels, low TH levels and chronic inflammation.
Response 3: Thank you for pointing this out, we have now correctly clarify the hormonal abnormalities that explain the relationship between these two diseases. See Abstract, page 1, line 14.
Point 4: Line15-16, Is the sentence correct? I think the authors want to mention that activity of THR-beta is effective in treating NAFLD.
Response 4: Thank you for bringing to our attention that the sentence was not sufficiently clear. We meant to say that the activity of THR-beta is paramount in the pathophysiology of hypothyroidism induced NAFLD, leading to the motioning of Resmetirom as possible treatment. We have, therefore, clarified this sentence. See page 1, lines 15-17.
Point 5: Line 44, What are "both diseases"?
Response 5: We agree, the sentence was not explicit enough, we have specified both diseases in the text. Thank you. See page 3, line 91.
Point 6: Line 46, "the effect that treatment..." might be "the effects of treatment..."?
Response 6: Thank you, we have changed the phrasing on the sentence to make it clearer. See page 2, line 93.
Point 7: Line 53, What is "this"? Hepatic lipogenesis or gluconeogenesis?
Response 7: Thank you for the observation. We agreed that the wording was incorrect. All the described actions are possible through those 2 mechanisms, so we have rewritten the sentence to reflect that. See page 3, line 100.
Point 8: Line 58, "ventricles," is "ventricles."?
Response 8: Thank you. In this sentence, we are indeed referring to heart ventricles. After reviewing the text, we find the word to be adequately written. Line 105.
Point 9: Figure1 is too busy to understand. I hope the authors to split into 2 figures, including an image of high TSH-low TH-chronic hepatic inflammation axis and a scheme of beneficial effects of THR-be-ta antagonist on NAFLD. And write down how to understand the Figure in legend. Line 64-78, there are other contents than Figure 1. See page 3, line 103.
Response 9: Thank you for this observation. We have divided the information into 2 figures. Figure 1 shows the relationship between hypothyroidism (high TSH and low TH) and liver steatosis. Figure 2 shows the beneficial effects of TRH-beta agonists. See Figure 1, page 2 and Figure 2, page 7.
Point 10: Line107, "summery" is "summary".
Response 10: Agreed. We have corrected this error. See page 3, line 161.
Point 11: From Line 110, there are explanations of DIOs except of DIO2. Is the information of DIO2 rare?
Response 11: Thank you for this interesting observation. There is indeed less information on DIO2. But there are important points regarding it, so we added an extra paragraph to further information on this topic. See page 4, line 238-240.
Point 12: Line 181-182, Is "such as diacylglycerols and ceramides" necessary here?
Response 12: Thank you. We agree that the sentence wasn’t needed, and it has been erased. See page 5, line 301.
Point 13: Line 270, "heigh" is "high-".
Response 13: Agreed. Thank you we have replaced the word. See page 6. Line 392.
Point 14: In Table1, It is better to unify the name of the compounds, whether it is a trade name or a generic name.
Response 14: Thank you for bringing this forward. We agreed and therefore, have used the compound generic name in all the examples in Table 1, to unify the text. See Table 1, page 7.
Round 2
Reviewer 1 Report
I have no further comments.
English is basically fine